# Additive Manufacturing of High-Entropy Alloys: A Review

**DOI:** 10.3390/e20120937

**Published:** 2018-12-06

**Authors:** Shuying Chen, Yang Tong, Peter K. Liaw

**Affiliations:** Department of Materials Science and Engineering, The University of Tennessee, Knoxville, TN 37909, USA

**Keywords:** high-entropy alloys, additive manufacturing, microstructure, mechanical properties

## Abstract

Owing to the reduced defects, low cost, and high efficiency, the additive manufacturing (AM) technique has attracted increasingly attention and has been applied in high-entropy alloys (HEAs) in recent years. It was found that AM-processed HEAs possess an optimized microstructure and improved mechanical properties. However, no report has been proposed to review the application of the AM method in preparing bulk HEAs. Hence, it is necessary to introduce AM-processed HEAs in terms of applications, microstructures, mechanical properties, and challenges to provide readers with fundamental understanding. Specifically, we reviewed (1) the application of AM methods in the fabrication of HEAs and (2) the post-heat treatment effect on the microstructural evolution and mechanical properties. Compared with the casting counterparts, AM-HEAs were found to have a superior yield strength and ductility as a consequence of the fine microstructure formed during the rapid solidification in the fabrication process. The post-treatment, such as high isostatic pressing (HIP), can further enhance their properties by removing the existing fabrication defects and residual stress in the AM-HEAs. Furthermore, the mechanical properties can be tuned by either reducing the pre-heating temperature to hinder the phase partitioning or modifying the composition of the HEA to stabilize the solid-solution phase or ductile intermetallic phase in AM materials. Moreover, the processing parameters, fabrication orientation, and scanning method can be optimized to further improve the mechanical performance of the as-built-HEAs.

## 1. Introduction

High-entropy alloys (HEAs) and multi-principal-element (MPE) alloys were proposed by Yeh [1] and Cantor [2] in the 2000s, respectively, attracting increasing interest all over the world. It was originally defined as an alloy composed of five or more multi-principal elements, with equi- or near equi-atomic percentages [1]. However, recently, researchers extended the concept of HEAs, which now include alloys with three or four principal elements as well [3]. Usually, they have a single crystal structure, such as a body-centered-cubic (BCC) [4,5,6], face-centered cubic (FCC) [7,8,9,10,11], or hexagonal-closed packed (HCP) structure [12,13,14,15]. HEAs present superior properties, such as a combination of high yield strength and ductility [16], good microstructural stability and retained mechanical strength at elevated temperatures [17,18,19,20,21,22,23], strong resistance to wear [24,25], fatigue [26,27,28,29,30,31], corrosion, and oxidation [32,33,34,35,36]. 

Most previous studies have focused on cast materials and their microstructure tuning through different post-processing methods, such as cold rolling, forging, or annealing treatment [37]. However, casting defects, such as shrinkages and pores, exist in as-cast materials, thereby requiring further processing to remove these defects. Compared with the conventional up–down fabrication method, additive manufacturing (AM), a flexible processing technique, has been applied to the fabrication of HEAs to produce materials with a complex geometry. AM, also known as three-dimensional (3D) printing, enables the fabrication of 3D objects based upon computer-aided design (CAD) models, which have been accepted as a transformative technology across multiple industries [38,39]. The AM technique has been becoming increasingly important in the materials science field and has been broadly applied in the industry for manufacturing products with complex shapes. Currently, several engineering materials can be produced by AM, such as aircraft components made of Ti alloys [40,41], Al alloys [42], stainless steel [43], and Polyamide 12 [44], which can significantly increase production efficiency and decrease the production cost due to the combined advantage of the net-shaping capability and design freedom [45]. Especially, several methods based on the AM concept were developed and commonly used in manufacturing products. They are classified as the laser metal deposition (LMD), selective laser melting (SLM, also called laser beam melting, LBM), laser metal fusion (LMF), direct metal laser sintering (DMLS) [46], or selective electron beam melting (SEBM) [39]. Direct metal deposition (DMD) and direct laser fabrication (DLF) are two representative LMD processes. 

LMD is characterized by the part being cladded layer by layer [47], as shown in Figure 1. The powders are melted after being carried by the inert gas into a laser beam and are then fed onto the workpiece to fuse with the thin layer deposited previously [47]. LMD can produce the 3D product with the ultrafine microstructure and highly complex geometry based on the layer-by-layer incremental shaping and consolidation of the feedstock to a wide range of configurations. The composition of the feeding powder could be modified by in-situ alloying during LMD, e.g., alloying varied amounts of Al to a CoCrFeMnNi powder blend, which will lead to even higher flexibility and throughput production [48,49]. 

SEBM is a melting fabrication process by a high-power electron beam in vacuum. Metal powders are normally fed from a hopper and then distributed by a rake across a build plate. A powder layer, which is uniformly supplied on the base plate, is pre-heated by the electron beam raster scanning, as shown in Figure 2. The points built in the slice data obtained from 3D-CAD file are sequentially melted by the focused electron beam. The process of powder feeding, pre-heating, and melting will be repeated until the bulk metallic parts is completed [51]. SEBM has attracted increasingly attention in the past decade due to its unique advantages of the high energy density of the high scan speed, incident electron beam, and reduced operation cost, which make it a suitable method to produce materials used the harsh environment, such as titanium alloys, aluminum alloys, cobalt chromium alloys, and nickel-based alloys [52].

SLM is an AM technology applying a high-energy laser beam, by which the part is fabricated in a layer-by-layer mode through the selective melting and consolidation of the metal powder [54], as shown in Figure 3. The layer thicknesses vary in a range of 20 and 100 μm. Compared with the traditional casting and forging method, SLM attracts increasing attention due to its impressive features, such as the ability to net-shape manufacture without the dies and high geometry complexity [55]. 

AM has gained increasing attention ascribed to its ability of producing parts with complex shapes. The laser-melting method has been used to HEAs to obtain coatings and bulk materials [57,58]. Brif et al. [55] studied the FeCoCrNi HEA alloy by the powder-bed technology, suggesting that the alloy remained in its single-phase solid-solution state and displayed excellent strength and ductility. Similarly, the phase evolution and mechanical properties of AlCoCrFeNi [52] and CoCrFeMnNi [59] HEAs were reported. In the current study, we review the application of AM in HEAs in terms of their microstructures, mechanical properties, and deformation mechanisms and compare them with conventional cast-HEAs. 

## 2. Microstructure Evolution during the AM Process

### 2.1. CoCrFeNi HEAs

Most of the previous studies are the HEA coatings [60,61]. Here, we highlight the performance of AM applications on bulk HEAs. The CoCrFeNi alloy system has been studied most, and more compositions or systems have been developed based on this alloy. For instance, Al*_x_*CoCrFeNi and CoCrFeNiMn HEAs have been commonly investigated in recent years. Brif et al. [55] studied FeCoCrNi HEAs fabricated by SLM, followed by annealing treatment. X-ray diffraction (XRD) results showed a single BCC solid solution with a uniform chemical composition. No segregation could be found. Later, Karthik et al. [62] proposed a new concept of the metal–metal composite, which consists of an aluminum–magnesium alloy, the AA5083 matrix, and the nanocrystalline CoCrFeNi HEA reinforcement precipitates in 12 vol%. Scanning electron microscope (SEM) images presented a very uniform distribution of HEA particles across the layers in a multi-layer composite. No intermetallic compounds, severe deformation, or accumulation of the HEA particles could be found at the interfaces, which was further confirmed by transmission electron microscope (TEM) examination. Furthermore, the TEM results presented a dynamic recrystallized aluminum matrix with fine equiaxed grains, which illustrated various dislocation densities, and some of the grains exhibited cells and subgrains in their formative processes. 

In order to improve their mechanical performance, Zhou et al. [63] obtained the C-containing FeCoCrNi HEAs fabricated by SLM with varying processing parameters. A single-FCC solid solution without the carbide phase was detected, and a uniform distribution of carbon in the matrix was found as well. Equiaxed grains existed mostly in the middle and bottom parts of the sample, and most grains in the top part were columnar. The geometry of grains in the SLM samples was found to be related to the cooling rate and thermal gradient. The electron-backscattered diffraction (EBSD) results revealed that the grain size and microstructure of the SLM specimens were highly dependent upon the laser power and scanning speed. For example, the specimen with a low power presented a larger number of equiaxed grains, and grains became irregular by increasing the scanning speed. 

### 2.2. Al_x_CoCrFeNi HEAs

Based on the CoCrFeNi composition, more AM investigations were performed on both composites and simple HEAs. For instance, with the addition of Al to CorCrFeNi, various microstructures were obtained by different processes. A single FCC phase in DLF Al_0.3_CoCrFeNi alloys was observed by Joseph et al. [64]. A large grain structure was found to be parallel and transverse to the build direction, with a strong texture of <001>, which could be due to the extensional growth of the material along the orientation of the deposition caused by the quick cooling rate and large thermal gradient in the molten pool. It should be noted that a small amount of very fine Ni- and Al-rich particles could be detected at grain boundaries. 

By increasing the content of Al to CoCrFeNi, Fujieda et al. [51] reported the BCC and FCC structures in AlCoCrFeNi after SEBM. Further EBSD results showed that the BCC crystal grew along the direction of <100>, which is a preferred orientation for crystal growth in the BCC alloy, which is along the build direction, coinciding with the heat-flux direction. Similarly, Shiratori et al. [65] prepared the AlCoCrFeNi by SEBM as well. More complex solid solutions of B2/BCC and FCC phases were obtained, which is different from the SEBM-specimen with BCC and FCC phases and the casting specimen with B2/BCC phases. Moreover, the confirmed Al–Ni-rich B2/Cr–Fe-rich BCC phases were found to be oriented along the build direction. The detailed elemental distribution in each phase is shown in Figure 4. An AlCoCrFeNi HEA with BCC, B2, and FCC phases along the grain boundaries was fabricated by Kuwabara et al. [52] by applying the SEBM method. Similarly, the SEBM samples presented fine and columnar grains along the building direction with a texture of <100>.

Li et al. [49] further investigated the properties of the ultrafine nanocrystals (UNs)-modified FeCoCrAlCu high-entropy alloy composites (HEACs) fabricated by the LMD, which presented the fine microstructure without the micro-crack with many AlCu_2_Zr UNs attached to the HEAC matrix. The lattice distortion was observed as well, due to the ultrafine microstructure and high diffusion of UNs destroying the atomic-equilibrium state, which increased the potential/free energy, thus leading to the formation of the lattice distortion. On the one hand, compression stress resulting from the alloying elements of the small atomic radius was generated. On the other hand, tensile stress resulting from the alloying elements of the large atomic radius was also achieved. The reciprocal interaction of these two kinds of stress fields led to decreasing stress and the formation of a relatively stable atomic group, favoring the formation of UNs. 

Niu et al. [66] investigated the phase evolution with varying the volumetric-energy density (VED), as shown in Figure 5. During the SLM process, the phase is mainly composed of A2 and B2 phases. Specifically, the B2 phase was mostly found to be distributed on the boundary of the molten pool, indicating that the B2 phase was the original structure due to the strongest combination between Al and Ni elements. Thus, the other elements were homogeneously dispersed around the Al–Ni B2 phase and formed the A2 phase. As the VED increased, the B2 phase increased, while the A2 phase tended to decrease, which was mainly due to the larger VED, inducing a faster cooling rate and thus leading to more B2 phases [66].

Sistla et al. [67] studied the Al/Ni ratio effect on the microstructure in Al*_x_*FeCoCrNi_2−*x*_ (*x* = 0.3 and 1) HEAs fabricated by the LMD method. The XRD and SEM results suggest that the studied solid solution transforms from BCC to FCC structures, which is consistent with the casting HEAs [67]. Joseph et al. [68] firstly proposed the HIP effect on the microstructure and mechanical behaviors in DLF Al*_x_*CoCrFeNi HEAs. The FCC, duplex FCC + BCC, and BCC solid solution were found in Al_0.3_CoCrFeNi, Al_0.6_CoCrFeNi, and Al_0.85_CoCrFeNi HEAs, respectively, which was similar to the casting specimens in previous reports [69]. It turned out that the HIP process removed all second-phase grain-boundary phases and segregation of the Al_0.3_CoCrFeNi alloy, indicating that the isothermal holding at 1100 °C for 2 h during HIP leads to the chemical homogenization of the material and the effectively dissolving of the grain-boundary phases, even though the HIP may induce microstructural coarsening.

### 2.3. CoCrFeMnNi HEAs

With the development of the AM research on HEAs, Cantor’s alloys have been employed in 3D printing as well. Haase et al. [48] investigated the 3D printing of the elemental powder blend in the CoCrFeMnNi system, producing elongated features with cellular dendrite structures. In fact, the fast cooling rates during the fabrication process could contribute to the formation of a fine cellular dendrite structure. The EDX-elemental mapping suggested a more homogeneous distribution of five principal elements than that in the casting specimens. Li et al. [70] firstly studied the processability of the non-equilibrium microstructure in the SLM CoCrFeMnNi. By increasing the VED, the microstructure became much denser. They showed that the HIP process removed the microcracks, and most micropores were closed, leading to an increased density, large grains, and a more homogeneous elemental distribution. The non-equilibrium processing of SLM resulted in a greater residual-stress difference than that in the HIP specimens. The TEM examination revealed the existence of an original FCC phase and a tetragonal-precipitation phase, which were due to the ultrafine grains and a large amount of dislocations induced in the SLM process. Moreover, nanotwins were found during the SLM process without plastic deformation, which may have been due to the low stacking fault energy caused by rapid solidification.

Guo et al. [71] firstly proposed the post-machining on the SLM CoCrFeMnNi HEA specimens to investigate the machinability. SEM illustrated that the mechanical polishing led to a uniform elemental distribution and smooth surface without clear waviness even though some microcracks and very small pores were present. Piglione et al. [59] proposed the printability of a CoCrFeMnNi HEA with single-layer and multi-layer builds fabricated by the laser-powder-bed fusion. A homogeneous distribution of composition was found in the bulk of the HEA with a single FCC crystal structure and a high degree of consolidation, without the apparent elemental segregation. In the single layer, the high cooling rate resulted in much finer cells, compared with previous reports [72] with the same composition. The high hardness values may be due to the restricted moving dislocation by cell boundaries during plastic deformation. Moreover, the cells were found to be aligned along the <001> orientation because the preferred growth direction is aligned with maximum heat flux. Similar cells with a reduced number could be found in multi-layers. The alternating sequence of columnar grains dominated by two crystallographic orientations was reported, which was caused by the coupling of the extensional growth and grain selection after remelting. 

In order to explore the AM materials application in a harsh environment, Qiu et al. [57] firstly reported the deformation mechanism at cryogenic temperature in the CrMnFeCoNi HEA fabricated by the LAM process. A single FCC crystal structure with a typical dendrite structure and a lattice constant of 3.598 Å were observed. The growth direction of the dendrite was found to be perpendicular to the laser-scanning direction due to the quick directional solidification. Zhu et al. [45] investigated the hierarchical microstructure in the CoCrFeNiMn HEA by SLM. The as-built specimens were obtained by controlling the processing parameters of the laser energy density, laser power, and scanning speed. The SEM images revealed that the columnar grains formed along the building direction (BD), with a <001> texture, implying epitaxial growth, which indicates that the formation of the cellular structure was correlated solidification conditions. A sub-micron cellular structure was detected under TEM, which was displayed with a large amount of dislocations and with clean interiors. No obvious segregation could be found from EDS, which was due to a fast cooling rate kinetically suppressing the segregation. After a heat treatment (HT), the cellular structure vanished with a much lower dislocation density left, suggesting their thermodynamically metastable characteristic [45]. Later, Fujieda et al. [73] investigated the SEBM method to prepare the CoCrFeNiTi-based HEA. The XRD results presented Ni_3_Ti intermetallic compounds with a uniform distribution in the matrix, different from the segregation in casting specimens, as shown in Figure 6. The needle-like NiTi precipitates with a basket-weave morphology contributed to the high yield strength in this SEBM specimen. 

### 2.4. Ti_25_Zr_50_Nb_0_Ta_25_ HEAs

The Ti_25_Zr_50_Nb_0_Ta_25_ refractory HEA fabricated by laser-melt deposition has been recently reported. Phase separation was found in the specimen, as shown in Figure 7a. The segregation near the grain boundary was found to be enriched in Ta and depleted in other elements, which was confirmed as a BCC crystal structure, similar to the matrix but with different compositions, shown in Figure 7b. Moreover, it was suggested that the grain size during the fabrication process was more likely influenced by the chemical partitioning of Ta in Zr-rich areas rather than the cooling rate [74]. The other MoNbTaW refractory alloy was prepared by the DMD method, but an additional remelting procedure between the powder deposition was required due to the different melting points of elements. The resultant compositional homogeneity of alloys was improved by preheating, laser parameters, geometries, and heat conduction [75]. 

## 3. Mechanical Properties of AM-Processed HEAs 

### 3.1. CoCrFeNi HEAs

Brif et al. [55] studied the mechanical performance of FeCoCrNi prepared by SLM and compared it with a more traditional processing route of arc melting, as shown in Figure 8. It was apparent that the AM specimen presented a much higher yield strength than that in the arc-melt specimen, while still retaining a significant portion of ductility. The enhanced mechanical properties of AM specimens were ascribed to a fine microstructure, due to the large temperature gradients and rapid solidification in SLM.

Karthik et al. [62] investigated the mechanical properties of the aluminum-magnesium alloy, an AA5083 matrix reinforced with 12 vol.% of CoCrFeNi HEA nanoparticles, i.e., the HEAp/5083 composite, fabricated by friction deposition. The composite demonstrated a high strength, resulting from the reinforcement precipitates of nanocrystalline CoCrFeNi, with a uniform distribution in the ultra-fine-grained aluminum matrix. It was suggested that the strengthening mechanism in the single-layer/HEAp/5083 composite could be the existing load transfer from the Al matrix to the reinforcement precipitation. The fracture morphology also revealed fine HEA reinforcement particles inside many of the dimples. The encouraging mechanical properties in the HEAp/5083 composite were due to the deficiency of brittle intermetallics at the particle/matrix and layer interfaces. The influence of SLM-processing parameters on mechanical behavior was studied in other C-containing FeCoCrNi alloys [63]. It was found that the yield strength was not significantly affected by the scanning speed and laser power. By applying the highest scanning speed of 1200 mm/s and a laser power of 400 W, the highest yield strength with a comparable ductility was obtained. It was apparent that the addition of carbon resulted in enhanced strength for FeCoCrNi, due to solid-solution strengthening. Work hardening was suggested to be caused by the interaction of dislocation–dislocation and dislocation–cellular walls since there was no deformation twinning. 

### 3.2. Al_x_CoCrFeNi HEAs

The mechanical properties of the addition of Al to CoCrFeNi alloys were extensively investigated in the AM HEAs. Joseph et al. [64] studied the asymmetry of tension/compression in the Al_0.3_CoCrFeNi alloy with a strong preferred orientation, fabricated by the DLF method. A significant difference in work hardening after yielding could be found in compression and tension curves, even though they had similar yield strengths. The limited work hardening in the tensile experiment was caused by cracks propagating along the grain boundaries, which may have been due to the precipitates enriched in nickel and Al, developing along the grain boundaries, which was observed in the laser-fabricated Rene88DT superalloy as well. Interestingly, deformation twins could be found in compression specimens, rather than tension specimens, because the stress required for the deformation twin could not be reached in the tensile sample before the final fracture. It was suggested that the loading axis in the tensile and compressive specimens were parallel to the depositing direction, which had a significant regulation with the texture of <001>, and was well arranged for deformation twins in the compressive experiment, but failed to orient for twinning in the tension experiment, due to the fact that only in certain directions can the polar deformation twinning accommodate the shape change. Thus, a high working-hardening rate was found in the preferred direction that favors twinning in the compression sample, revealing that the strong texture in the primary material integrated with the activation of twinning contributed to the difference in work-hardening behavior in the tensile and compressive experiments. 

With the increase in Al content, the mechanical properties in the AlCoCrFeNi alloy was investigated by the SEBM method [65]. The compressive stress–strain curve presented much better plasticity in SEBM specimens than that in the cast specimens [51,52], due to their finer grains obtained at a higher cooling rate. The cooling rate in the solidification process during the SEBM varied in the range of 10^−3^–10^−5^/s, which was much greater than the conventional casting case even with a water-cooled copper mold. This trend could be a reason for the improved deformability in SEBM alloys. Moreover, the generation of the FCC structure could contribute to the good ductility in SEBM materials. It was concluded that the cast specimen of AlCoCrFeNi almost solely consisted of the B2 and BCC phase mixture, which lacks the slip system and leads to a brittle fracture. However, in SEBM specimens, a substantial fraction of the ductile FCC phase was found, which will favor the multi-slip system and thus enhance ductility during the compression experiment, as shown in Figure 9. 

Similar work and results were obtained in AlCoCrFeNi alloys [51,52] as well. Moreover, the anisotropy of the compressive specimen was observed, which was ascribed to a large number of grain boundaries [51]. The SEBM sample along the BDs exhibited improved properties, but the reduced yield strength and plasticity were found in samples perpendicular to the build direction (BD). Moreover, their compressive properties were much higher than the tensile properties, which was consistent with previous studies [68]. As previously discussed, FCC and B2 ordered phases existed in the SEBM materials. The fracture of the tensile specimens was ascribed to the preferential formation and propagation of the cracks along BCC/FCC boundaries induced by the significant difference in the elastic properties of these adjacent phases. The micro-voids found in the failed specimen were attributed to the shrinkage induced by the denser FCC phases than BCC or B2 structures [45]. Li et al. [49] proposed the mechanical performance of the composite of the FeCoCrAlCu alloy deposited on the titanium alloy via the LMD method. The excellent wear resistance of materials was primarily attributed to their multiple structure, such as the quasi-crystalline/nanocrystalline phases, and the free micro-crack microstructure. The enhanced strength and ductility induced by adding the proper amount of Y_2_O_3_ content could lead to the improvement of the wear resistance. Moreover, the spreading nanoscale particles were capable of withstanding the external normal load, leading to increased wear performance. 

### 3.3. CoCrFeMnNi HEAs

The mechanical properties and formability of CoCrFeMnNi HEAs were widely investigated by various AM methods [41,42,43,44,45,46,47,48,49,50,51]. Haase et al. [48] revealed the high formability and significant high hardness of CoCrFeMnNi alloys fabricated by the LMD method, producing excellent yield strength and ductility. It was clear that the yield strength of LMD-produced materials presented a much higher value than that in cast materials, which may have been due to the pronounced texture, lowering the mean Schmid factor, thus requiring higher stress to initiate dislocation motion. Furthermore, the initial high density of dislocation caused by the fast solidification and cooling in the LMD process contributed to the increased yield strength as well. 

Li et al. [70] studied the mechanical properties of the CoCrFeMnNi HEA by the SLM method. Holes and cracks were reduced by increasing the VED, leading to higher ultimate tensile strength. The tensile strength was improved by HIP, which was ascribed to the closure of micro-pores and micro-cracks, accompanied by the reduced elongation. Overall, the SLM specimen presented a higher tensile strength than that in the slowly solidified HEA, which could be explained in terms of the Hall–Petch theory, which shows that decreasing the grain size leads to high strength. Moreover, the precipitation of the σ phase, preserving from high temperatures owing to rapid solidification, contributed to the improvement of the tensile strength. Furthermore, a similar investigation was performed on the strengthening mechanism in an SLM CoCrFeNiMn HEA with hierarchical microstructures [45]. All the SLM specimens presented much better mechanical properties than that in the as-cast specimens. The postmortem STEM indicated a significant dislocation arrest and retainment mechanism within the deformation cells, resulting in a distinct increase in the dislocation density within the cell walls, which is consistent with Piglione’s [59] work. Moreover, the planar sliding interact with the cellular structure substantially for forming a 3D dislocation network, which dominates the deformation process in the as-built HEAs, accompanied by the additional contribution from deformation twinning. Based on the calculation of the yield strength dependent on the dislocation density, they induced that the improved strength was mostly controlled not only by grain-boundary strengthening and friction stress but also by dislocation hardening. Furthermore, the excellent uniform elongation of the as-built HEA was due to the capability of steady strain-hardening at large stress levels. Specifically, dislocations being significantly arrested and retained in the cell interior, and substantial interactions between slip bands and cellular structures, resulted in the generation of a complicated dislocation configuration, which led to a comparable strain-hardening ability. 

Another way of studying LAM-fabricated HEAs was investigated to explore the deformation mechanisms at cryogenic temperatures [57]. Their yield strength and ductility were enhanced when the temperature decreased from 298 to 77 K, as shown in Figure 10. The strength in the LAM specimen was found to be higher than that in the as-cast sample, which was due to the suppressed elemental segregation and possible high dislocation density induced by the fast solidification rate in the LAM sample. Deformation twinning was found in the deformed sample. When increasing the plastic strain, the deformation twinning was more obvious. The increased deformation twinning was more dispersed as the strain reached the highest level. However, the deformation mechanism could not be dominated by deformation twinning even at high strain levels due to the limited quantity and its heterogeneous distribution. The local misorientation map indicated that the dislocation is the dominant deformation mechanism, especially with a low strain value. The density of dislocations increased rapidly and uniformly with increasing strain, which revealed that dislocations played an essential role during deformation. It was noticeable that the initial dislocation density in the LAM specimen was substantial, larger than that in the reported materials, which may have been due to rapid solidification and cooling in the LAM process. Such microstructures will undoubtedly produce an enhanced yield strength. It should be noted that, under the cryogenic condition, the deformation twinning at high strain levels could be an obstacle to moving dislocation, resulting in a continuous accumulation of dislocations, thus leading to a high work-hardening rate as well as an increased strength. 

Fujieda et al. [73] investigated the tensile properties of the CoCrFeNiTi-based HEA, achieved using SEBM and the subsequent solution treatment. The untreated SEBM sample presented a much higher tensile strength than that in the as-cast specimen, due to the uniform distribution of the needle-like Ni_3_Ti in the matrix, while a reduced ductility was found as well, which should be attributed to the cracks progressing along the boundaries between excessive Ni_3_Ti intermetallic compounds and the matrix. 

### 3.4. Post-Treatment Effect

It has been proposed that post-treatment has a significant effect on the mechanical properties of AM-processed HEAs. It is believed that defects such as residual stress may exist in a specimen during deposition, which may affect the mechanical properties. For example, after annealing at 750 °C, a CoCrFeNi specimen presented a reduced yield and tensile strength with a minor reduction in ductility. When annealing temperature increased to 1000 °C, this CoCrFeNi specimen presented a lower yield strength and tensile strength, but improved ductility, which was due to the stress relief and grain growth (shown in Figure 8) [55].

The HIP effects on mechanical properties were studied in the DLF Al*_x_*CoCrFeNi alloy (*x* = 0.3, 0.6, and 0.85) [68]. The DLF/HIP Al_0.3_CoCrFeNi presented an enhanced ductility and work-hardening rate, without losing its yield strength. However, the DLF/HIP Al^0.3^ showed an improved yield strength but reduced ductility. A small volume fraction of deformation twinnings was found in the tension-failed specimens of the DLF/HIP Al_0.3_CoCrFeNi, which was due to a lower stress compared with the critical stress to activate the twinning. The formation of the hard B2 grain-boundary phase by the HIP process was detrimental to tension, which led to reduced ductility. This result was proved by the fact that the B2 phase in Al*_x_*CoCrFeNi failed to accommodate shape changes. Sistla et al. [67] studied the heat-treatment effect on the Al*_x_*FeCoCrNi_2−*x*_ prepared by DMD. It was found that the Fe–Cr matrix in the as-deposited, annealed, and as-quenched specimen was enhanced by the precipitates of the Ni- and Al-rich phases. The Al–Ni composition will transform to a two-phase domain with NiAl + Ni_3_Al at higher temperatures, resulting in further improvment in the strength of materials. 

A SEBM–CoCrFeNiTi-based HEA exhibited good tensile properties [73]. In order to further improve the mechanical properties of SEBM materials, a solution treatment was performed, leading to the disappearance of Ni_3_Ti. Only small particles composed of Ni and Ti elements were present in the specimen. It is obvious that the ductility was improved without lowering their strengths after solution treatment (ST), as shown in Figure 11. The excellent ductility induced by the delay at the beginning of the plastic instability was attributed to the dynamic recovery induced by the screw-dislocations cross slips or the dislocations coalescence and annihilation during the plastic deformation. Specifically, the air-cooled (A.C.) specimen illustrated a higher yield strength but a lower ductility than that in the water-quenched (W.Q.) specimen, which was ascribed to the different precipitate size of the ordering phases that can act as a weak obstacle to the moving dislocation. The particle size of the sample obtained by the A.C. method was about 40 nm, higher than that of the W.C. sample, with a particle size of 10 nm. It was concluded that the moving dislocation was able to cut through the small particles with a diameter of several tens of nanometers. Moreover, the critical resolved shear stress increased proportionally to the square root of the precipitate diameter.

### 3.5. Comparison of Mechanical Properties

Figure 12 presents the yield strength and ductility obtained from both compression and tension experiments in HEAs, conventional Al, TiAl, and Cu alloys, and steels alloys fabricated by AM [60,76]. It is apparent that HEAs present a much higher yield strength and impressive plasticity than that in Al, TiAl, and Cu alloys and in steels. Even though the AlCoCrFeNi alloy fabricated by casting or by SEBM was found to have a high yield strength in the tensile tests, limited ductility was observed as well, which was due to the brittle phase appearance in the matrix. The modified Co_1.5_CeFeNi_1.5_Mo_0.1_ HEA fabricated by SEBM showed an excellent combination of yield strength and ductility in the tensile experiments due to the uniform dispersion of small Ni_3_Ti particles in the matrix. The CoCrFeNi alloys prepared by SLM demonstrated a higher yield strength than that in the LMD CoCrFeNi and CoCrFeNi HEA/5083 composite, but comparable to the SLM C-containing FeCoCrNi HEAs, which was ascribed to their fine microstructure in SLM materials. The LAM CoCrFeMnNi HEA suggested that the HIP-post process enhanced yield strength, but slightly lowered elongation. The SLM CoCrFeMnNi HEAs depending on various scanning speeds and heat treatments displayed a wide range of ductility and comparable yield strength. The DLF Al_0.3_CoCrFeNi alloys revealed a much lower yield strength, compared with other methods, as apparent in Figure 12. We can conclude that different AM methods induce substantial differences in both yield strength and plasticity, and the post-heat-treatment enhances their properties by removing various defects and releasing the residual stress present in the materials. 

## 4. Future Work

HEAs are potential materials that can act as functional and structural materials due to their excellent properties. The complex geometries of HEAs cannot be realized by the conventional method. Therefore, AM paves the way to the fabrication of HEAs with complicated shapes. However, defects are present in AM materials, which degrade mechanical properties. Thus, it is essential to explore ways to optimize the microstructure and enhance the mechanical properties of AM-processed HEAs. Suggestions are as follows:The residual stress can occur in as-built components during the high cooling process, which may cause crack growth. It is necessary to further optimize the processing parameters and heat treatment regime to improve the microstructures and mechanical properties.Most work highlights tension or compression experiments. Investigations on their fracture toughness, fatigue properties, oxidation, irradiation, or corrosion properties are rare. It is essential to study AM HEAs in a wide range of applications.Moreover, most reported alloys are Al_*x*_CoCrFeNi or CoCrFeMnNi systems. Little work has been performed on light-weight HEAs or refractory HEAs, which is another challenge for AM investigation in the future.It is essential to develop a broader compositional range and set of mechanical properties, which could be a guideline to properly choosing an appropriate set of reinforcement compositions and designing an AM product suitable for a complex service environment.Investigating materials with large differences in melting temperature is another challenge.The starting powder feature should have an important influence on the fabricated materials during the AM process. Hence, it is essential to investigate an effective way of blending the powder directly or indirectly to reduce agglomeration and increase homogeneity.

## 5. Conclusions

The application of AM in the fabrication of HEAs and the post-heat-treatment were reviewed. The improved mechanical behavior of AM samples, compared with casting materials, is ascribed to the refinement of microstructures caused by large temperature gradients, rapid solidification, and cooling in the fabrication process, which is of significant importance in highly alloyed systems. The post-heat-treatment enhances their properties by removing various defects and releasing the residual stress formed in AM materials. Furthermore, reducing the pre-heating temperature to prevent phase segregation or modifying the composition of the HEA to stabilize one of the phases of BCC, FCC, and B2 structures were considered to improve the mechanical properties in AM materials. It is feasible to further improve the mechanical performance of as-built materials by optimizing the processing parameters, the scanning method, and the fabrication orientation. Promising results pave the way to investigate HEAs as engineering materials by AM methods. 

## Figures and Tables

**Figure 1 entropy-20-00937-f001:**
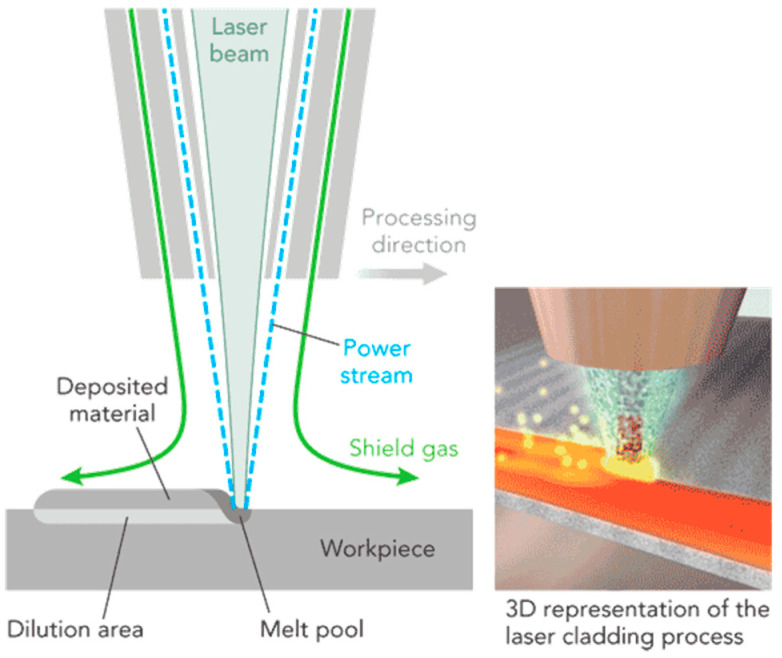
Schematic image of LMD. Image displayed with the permission from the authors in [50].

**Figure 2 entropy-20-00937-f002:**
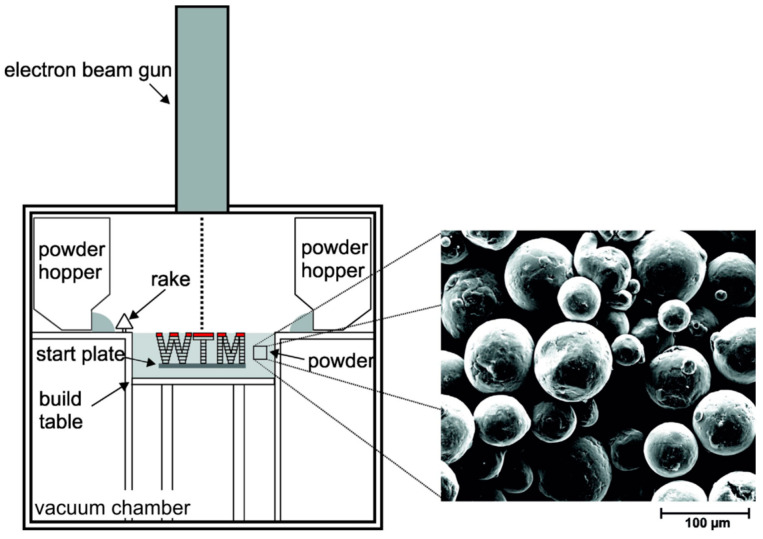
Schematic image of the SEBM process. Image displayed with permission from the authors in [53].

**Figure 3 entropy-20-00937-f003:**
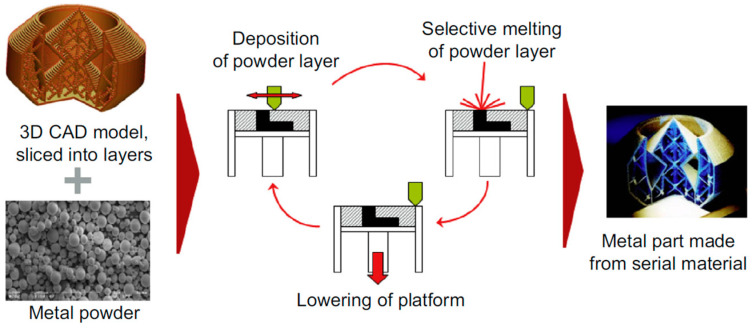
Schematic image of the SLM process. Image displayed with permission from the authors in [56].

**Figure 4 entropy-20-00937-f004:**
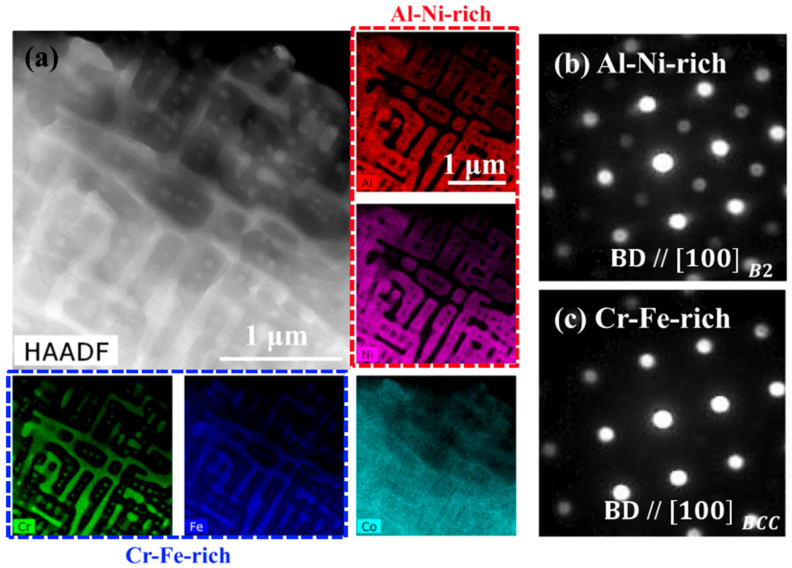
(**a**) The high-angle annular dark-field scanning transmission electron microscopy (HAADF–STEM) image and elemental maps obtained from the energy-dispersive X-ray spectroscopy (EDX). (**b**,**c**) Nano-beam diffraction (NBD) patterns obtained from the Al–Ni-rich and Cr–Fe-rich regions, respectively. Image displayed with permission from the authors in [65].

**Figure 5 entropy-20-00937-f005:**
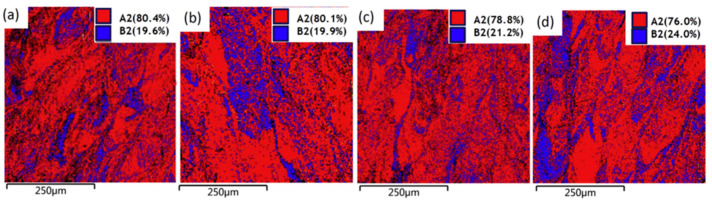
Phase dispersion of different VEDs in SLM samples: (**a**) 68.4 J/mm^3^; (**b**) 83.3 J/mm^3^; (**c**) 97.2 J/mm^3^; and (**d**) 111.1 J/mm^3^. Image displayed with permission from the authors in [66].

**Figure 6 entropy-20-00937-f006:**
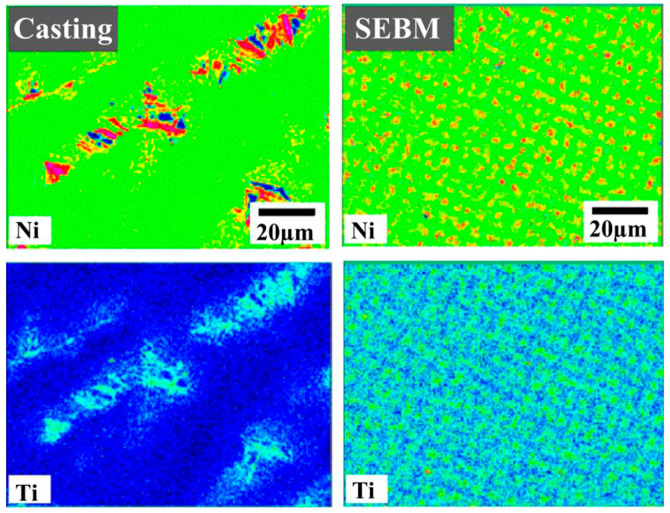
EPMA elemental maps of Ni and Ti in the as-cast sample and SEBM sample. Image displayed with the permission from the authors in [73].

**Figure 7 entropy-20-00937-f007:**
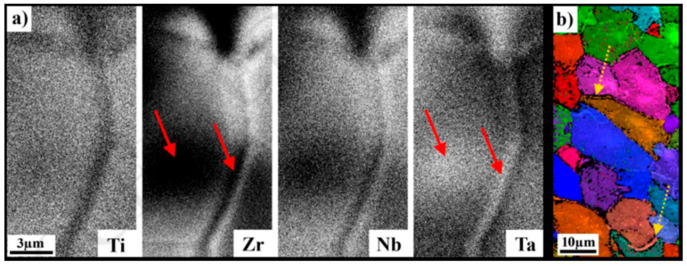
(**a**) Ta-enriched in the grain boundary and Ta-rich dendritic core inside the grain (illustrated by red arrows). Nb, Zr, and Ti concentrations showing an opposite trend (illustrated by red arrows). (**b**) EBSD presenting the grain-orientation map. Image displayed with permission from the authors in [74].

**Figure 8 entropy-20-00937-f008:**
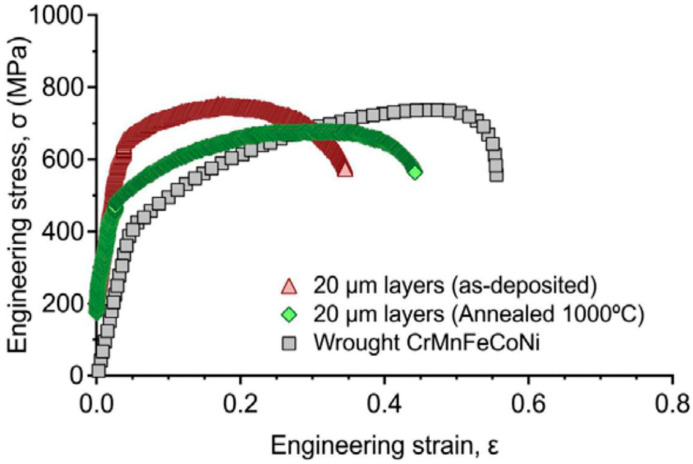
Representative tensile stress–strain curves of AM specimens with a 20-μm-layer thickness. Image displayed with permission from the authors in [55].

**Figure 9 entropy-20-00937-f009:**
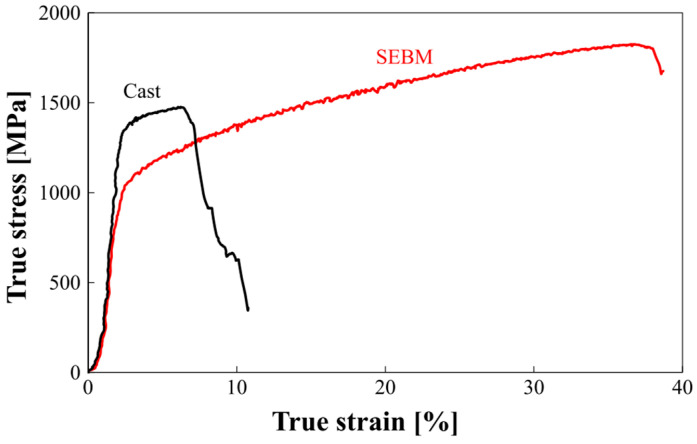
True stress–strain curves in the cast and SEBM samples. Image displayed with permission from the authors in [65].

**Figure 10 entropy-20-00937-f010:**
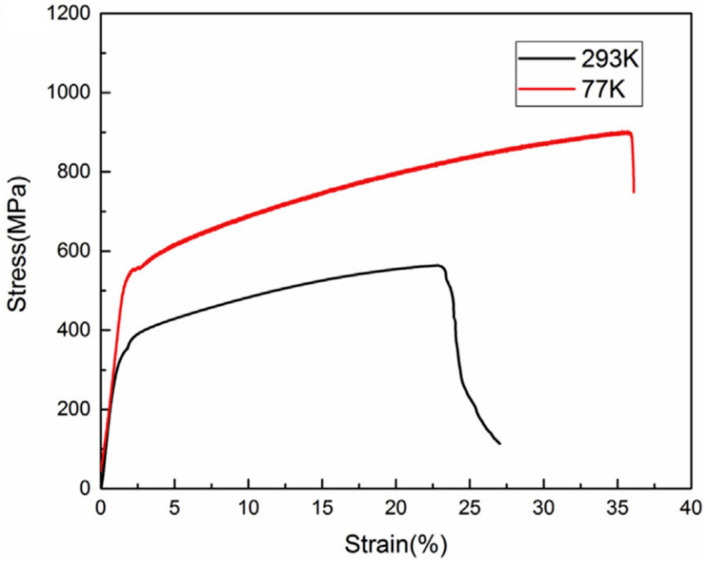
Tensile engineering stress–strain curves in the CoCrFeMnNi HEA tested at 293 K and 77 K, respectively. Image displayed with permission from the authors in [57].

**Figure 11 entropy-20-00937-f011:**
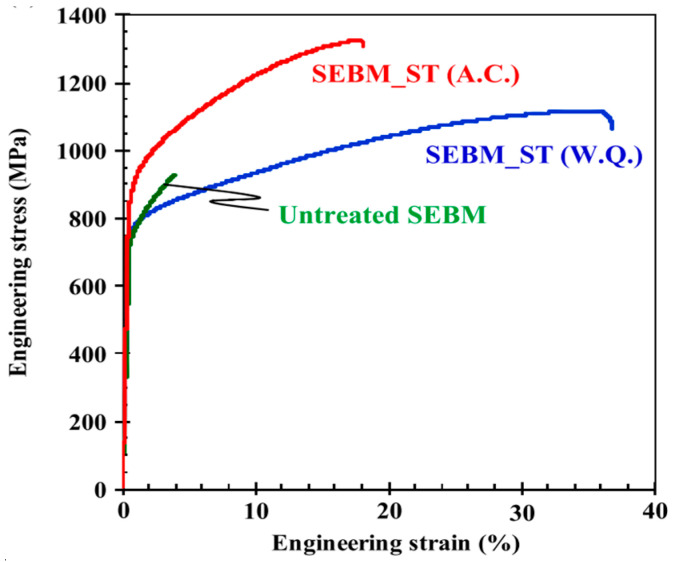
Engineering stress-engineering strain curves of deposited, and solution-treated SEBM specimens tested at room temperature. Image displayed with permission from the authors in [73].

**Figure 12 entropy-20-00937-f012:**
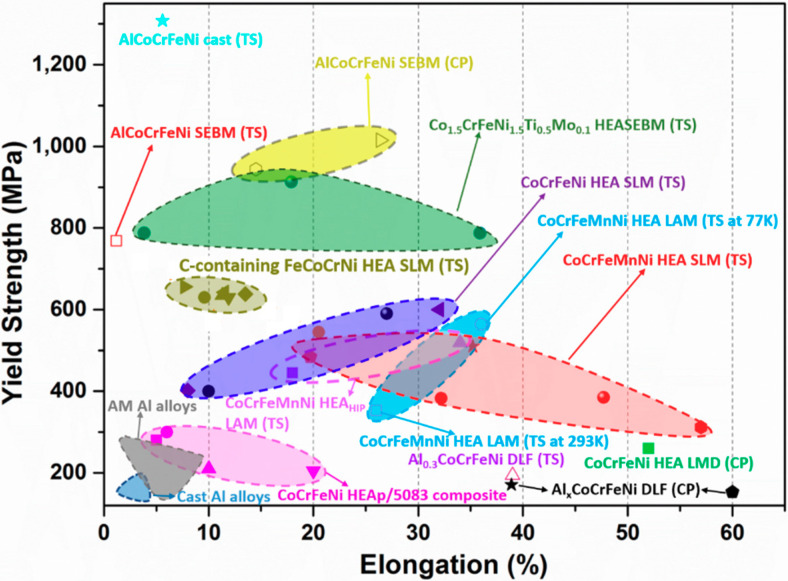
Yield strength vs. elongation using various AM methods. (TS: Tensile; CP: Compression).

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
