# Peer review of "Additive Manufacturing of High-Entropy Alloys: A Review"

_entropy, 2018, doi:10.3390/e20120937_

Round 1

Reviewer 1 Report

The MS is a review on the additive Manufacturing of High-Entropy Alloys. In general, the paper is well organizes and the topic is very interesting for readers of Entropy. On this basis, I recommend its publication addressing the following comments:

-  Figs. 2,13. The quality and readibility of these figures are very low. Please check and revise.

-  Recent articles (Entropy 2018, 20(9), 646; Materials 2018, 11(1), 53; Materials 2017, 10(11), 131; Metals 2017, 7(11), 482) on High-Entorpy Alloys should be mentionied in the Introduction.

Author Response

Reviewer 1

The MS is a review on the additive Manufacturing of High-Entropy Alloys. In general, the paper is well organizes and the topic is very interesting for readers of Entropy. On this basis, I recommend its publication addressing the following comments:

-  Figs. 2,13. The quality and readibility of these figures are very low. Please check and revise.

Reply: Thank you. They were corrected

-  Recent articles (Entropy 2018, 20(9), 646; Materials 2018, 11(1), 53; Materials 2017, 10(11), 131; Metals 2017, 7(11), 482) on High-Entropy Alloys should be mentioned in the Introduction.

Reply: Thank you. These suggested references were added.

Reviewer 2 Report

I think this review is interesting and is a valuble contribution both to the area of HEAs and to the area of AM.

I have read it with pleasure and learned new things from this review.

Here are my suggestions to improve the presentation quality of the material:

Please add a sentence in the beginning of the abstract to explain what stimulated the authors to write this review (for example, the areas in question are rapidly developing, no review has been written so far and the community needs a work describing the state of the art).

I am concerned with HIP being "high isostatic pressure" in the article. It seems that you meant "hot isostatic pressing". Please correct.

It seems that "AM" was not explained as abbreviation in the Introduction. 

Please edit Fig. 2, something is wrong with the notation layout (double notations appear).

If figures are adopted or re-printed, please provide not only the reference itself but the words "reprinted/reproduced with permission from ..." (the exact phrase is recommended by the publishers of the corresponding works).

Please correct Fig. 12 (y-axis) and Fig. 13 (the upper part, "compression" is only half seen").

I suggest that the authors rethink the title of section 3. Now it is "Discussion" and 3.4 comes as "Comparision of mechanical properties". Maybe section 3 should be entitled "Mechanical properties of AM-processed HEAs". Also, I suggest making a "post-treatment effect" separate subsection.

I think the "Future work" section should be somewhat extended to discuss why the directions that were mentioned should be pursued after all (not just that there has been little work on them).

Author Response

Reviewer 2

I think this review is interesting and is a valuable contribution both to the area of HEAs and to the area of AM.

I have read it with pleasure and learned new things from this review.

Here are my suggestions to improve the presentation quality of the material:

Please add a sentence in the beginning of the abstract to explain what stimulated the authors to write this review (for example, the areas in question are rapidly developing, no review has been written so far and the community needs a work describing the state of the art).

Reply: Thank you very much for your suggestion. The motivation was added in the abstract.

I am concerned with HIP being "high isostatic pressure" in the article. It seems that you meant "hot isostatic pressing". Please correct.

Reply: It was corrected

It seems that "AM" was not explained as abbreviation in the Introduction.

Reply: It was corrected

Please edit Fig. 2, something is wrong with the notation layout (double notations appear).

Reply: It was corrected

If figures are adopted or re-printed, please provide not only the reference itself but the words "reprinted/reproduced with permission from ..." (the exact phrase is recommended by the publishers of the corresponding works).

Reply: Thank you. They were corrected

Please correct Fig. 12 (y-axis) and Fig. 13 (the upper part, "compression" is only half seen").

Reply: It was corrected.

I suggest that the authors rethink the title of section 3. Now it is "Discussion" and 3.4 comes as "Comparision of mechanical properties". Maybe section 3 should be entitled "Mechanical properties of AM-processed HEAs". Also, I suggest making a "post-treatment effect" separate subsection.

Reply: Thank you. The title of section 3 has been change to “Mechanical properties of AM-processed HEAs” and the “Post-treatment effect” was separated as a subsection.

I think the "Future work" section should be somewhat extended to discuss why the directions that were mentioned should be pursued after all (not just that there has been little work on them).

Reply: Thank you for your suggestion. The future work section has been extended, shown in the text.
